# Thermoelectric Behavior of BaZr_0.9_Y_0.1_O_3−d_ Proton Conducting Electrolyte

**DOI:** 10.3390/membranes9090120

**Published:** 2019-09-13

**Authors:** Dmitry Tsvetkov, Ivan Ivanov, Dmitry Malyshkin, Vladimir Sereda, Andrey Zuev

**Affiliations:** 1Institute of Natural Sciences and Mathematics, Ural Federal University, Ekaterinburg 620000, Russia; Ivan.Ivanov@urfu.ru (I.I.); Dmitry.Malyshkin@urfu.ru (D.M.); Vladimir.Sereda@urfu.ru (V.S.); Andrey.Zuev@urfu.ru (A.Z.); 2Institute of High Temperature Electrochemistry, Ural Branch of Russian Academy of Sciences, Ekaterinburg 620000, Russia

**Keywords:** proton conductor, BaZr_0.9_Y_0.1_O_2.95_, seebeck coefficient, conductivity, hydration

## Abstract

BaZr_0.9_Y_0.1_O_3-δ_ (BZY10), a promising proton conducting material, exhibits p-type conduction under oxidative conditions. Holes in BZY10 are of the small polaron type. However, there is no clear understanding at which places in the lattice they are localized. The main objectives of this work were, therefore, to discuss the nature of electronic defects in BZY10 on the basis of the combined measurements of the thermo-EMF and conductivity. Total electrical conductivity and Seebeck coefficient of BZY10 were simultaneously studied depending on partial pressures of oxygen (*pO_2_*), water (*pH_2_O*) and temperature (*T*). The model equation for total conductivity and Seebeck coefficient derived on the basis of the proposed defect chemical approach was successfully fitted to the experimental data. Transference numbers of all the charge carriers in BZY10 were calculated. The heat of transport of oxide ions was found to be about one half the activation energy of their mobility, while that of protons was almost equal to the activation energy of their mobility. The results of the Seebeck coefficient modeling indicate that cation impurities, rather than oxygen sites, should be considered as a place of hole localization.

## 1. Introduction

Perovskite-type yttrium-doped barium zirconate BaZr_0.9_Y_0.1_O_3-δ_ (BZY10) is a well-known proton conducting material promising for intermediate-temperature electrochemical applications due to high proton conductivity and good chemical stability [1,2,3,4]. The defect chemistry of BZY10, which is of key importance for understanding its conductivity, was studied using a number of techniques such as thermogravimetry (TGA) in water vapour containing atmosphere [2,3,5,6], conductivity measurements under wide range of conditions [4,5,7], quasi-elastic neutron scattering (QENS) [8,9], nuclear magnetic resonance (NMR) [10], X-ray absorption spectroscopy (XAS) [11] and computer simulations [12,13,14,15]. As a result, the nature of proton defects in BZY10 is relatively well studied [16,17]. However, this is definitely not the case concerning electronic defects. Indeed, it is well known that under certain conditions BZY10 exhibits significant hole-type electronic conductivity comparable to, or even exceeding, the ionic one [5,6,7]. This p-type conductivity arises due to the partial filling of oxygen vacancies, created through Y-doping, by oxygen under an oxidizing atmosphere [18,19,20]. In our previous paper [21] we showed that holes in BZY10 are of the small polaron type and exhibit rather strong trapping by some acceptor impurity. Besides yttrium dopant cations, various 3d-metal cations, small amounts of which are always present in any material and are sometimes intentionally introduced during the synthesis as sintering aids [22,23,24], may act as a hole trap. So far, however, it is not clear on which positions in the lattice that the non-trapped holes are localized. The assumption that the holes are associated with the oxygen sites is consistent with the results of BZY10 electronic structure calculations [25] and in line with the conclusions of Schirmer and others [26] and references therein, in regards to the nature of holes in the great number of different oxide materials. However, the hypothesis that the holes are localized on the impurity ions is also quite plausible, taking into account the rather small concentration of holes present in BZY10. In this respect, it is worth noting that the thermoelectric behaviour of BaZr_1-x_Y_x_O_3-d_ proton conducting oxides has not been studied and discussed in detail so far, contrary to electrical conductivity, although thermoelectric power measurements were shown to provide valuable data necessary for elucidating the nature of defects and charge transfer in case of the state-of-the-art oxide ion conductors and mixed conductors [27,28,29,30,31,32,33,34]. Therefore, combining the thermo-EMF and conductivity measurements should advance the understanding of both defect chemistry and charge transfer in proton conductors.

Therefore, the main objectives of this work are (i) to study the thermoelectric behaviour of proton conducting oxide BZY10 and its electrical conductivity in the wide range of temperatures (*T*), oxygen (*p*O_2_) and water vapour (*p*H_2_O) partial pressures, and (ii) to discuss the nature of electronic defects in BZY10 in the light of these measurements.

## 2. Materials and Methods 

The powder sample of BZY10 was prepared by means of glycerol-nitrate method described elsewhere [15]. High-purity BaCO_3_ (99.99%, Lanhit, Moscow, Russia), Zr(OH)_2_CO_3_ (99.9%, ChMZ, Glazov, Russia) and Y_2_O_3_ (99.99%, Lanhit, Moscow, Russia) were used as starting materials. The BZY10 powder was calcined three times at 1100 °C in air. As-prepared powder was pressed at 154–160 MPa into tablets 20 mm in diameter and then sintered at 1500 °C for 24 h in air with 100 °C/h as a heating/cooling rate. Before the sintering procedure, green tablets were covered by the sacrificial powder of the same chemical composition in order to prevent Ba loss due to volatility of its oxide at high temperatures. The relatively moderate sintering temperature was chosen because of the same reason. Rectangular bars of 20 × 3 × 3 mm^3^ for the conductivity and Seebeck coefficient measurements were cut from the tablets sintered accordingly. The as-obtained bars had 85–90% relative density. The residual porosity promoted sample equilibration with the gas phase, as discussed in detail by Wang and Virkar [16]. This significantly reduced the time necessary for the sample equilibration and, consequently, allowed for obtaining reproducible equilibrium values of all the properties studied in this work.

The phase composition of the sample prepared accordingly was studied by means of X-ray diffraction (XRD) using an XRD 7000 diffractometer (Shimadzu, Kyoto, Japan). The XRD (Cu Kα radiation) showed no indication for the presence of a second phase in the as-prepared powder of BZY10.

Electrical conductivity was measured by a 4-probe dc method using an original homemade setup [21] equipped with an yttria-stabilized zirconia (YSZ) electrochemical oxygen pump and sensor. The Seebeck coefficient was measured simultaneously in the same setup by placing the sample in the moderate temperature gradient, around 10–15 °C. The uncertainties in the total conductivity and the Seebeck coefficient did not exceed ±5% and ±10 μV/K, respectively. The measurements were carried out depending on *T*, *p*O_2_ and *p*H_2_O.

Water uptake of as-prepared BZY10 was measured by the thermogravimetric technique (TG) [35]. The measurements were carried out in the temperature range 25–900 °C in the atmosphere of air (log(*p*O_2_/atm) = −0.677) or nitrogen (log(*p*O_2_/atm) = −4.5) and in the range of water vapor partial pressure (−1.73 ≤ log(*p*H_2_O/atm) ≤ −4.0).

The atmosphere employed in the thermo-EMF, conductivity and TG measurements was dried by circulating the gas through the column filled with pre-annealed zeolites. The residual water vapor pressure was found to be log(*p*H_2_O/atm) = −4.0. The humidity of the gas (i.e., *p*H_2_O) was measured using the original unit on the basis of the H_2_O-sensor BME-280 (Bosch, Stuttgard, Germany).

## 3. Results and Discussion

The XRD pattern of the as-prepared single-phase BZY10 was indexed using the cubic $Pm\bar{3}m$ space group and is shown in Figure 1 along with the Rietveld refinement results. The refined value of BZY10 lattice parameter, ***a*** = (4.2075 ± 0.0001) Å, is in good agreement with its values reported previously [1,2,3,4,5,7].

The total conductivity and Seebeck coefficient of BZY10 were measured as a function of *p*O_2_, *p*H_2_O and *T* in the ranges −18 ≤ log(*p*O_2_ /atm) ≤ −0.67, −4 ≤ log(*p*H_2_O /atm) ≤ −1.73 and 688 ≤ T (°C) ≤ 1038. As an example, the results obtained at log(*p*H_2_O /atm) = −1.73 are shown in Figure 2.

As seen, the total conductivity behavior is typical of many proton-conducting oxides, with a characteristic slope of 1/4 corresponding to a p-type contribution under oxidizing conditions and almost *p*O_2_-independent ionic part. The ionic contribution was found to possess the *p*H_2_O dependence with a characteristic slope of 1/2 under reducing atmosphere. At the same time, the Seebeck coefficient is positive and grows with decreasing *p*O_2_ under oxidizing atmosphere. Under reducing conditions (*p*O_2_ < 10^−5^), it drops abruptly, becoming negative, and increases upon subsequent *p*O_2_ decrease. It is also to be emphasized that within the *p*O_2_ range of 10^−10^–10^−5^ atm it was impossible to obtain stable values of Seebeck coefficient. The reason for both the abrupt change and the instability of Seebeck coefficient mentioned above is most probably related to the fact that the heterogeneous part of the thermo-EMF is strongly dependent on the chemical potential of oxygen in the surrounding atmosphere [27,31,33,34]. The chemical potential, in turn, is determined by gaseous O_2_ under oxidizing conditions (10^−5^ ≤ *p*O_2_ ≤ 0.21 atm) and by the H_2_/H_2_O chemical equilibrium under reducing atmosphere (at *pO_2_* lower then ~10^−10^ atm). Thus, it is difficult to achieve stable and well-defined chemical potential of oxygen in the intermediate pO_2_ range and, consequently, it is also practically impossible to get reliable values of the Seebeck coefficient. Therefore, in order to analyze the behaviour of sample’s thermoelectric power and to see its overall trend vs. *p*O_2_, one needs to normalize the Seebeck coefficient measured in H_2_/H_2_O atmosphere against that obtained under the O_2_/N_2_ atmosphere. It can be easily shown [27,31,33,34] that the difference between the coefficients measured at the same *p*O_2_ under O_2_/N_2_ atmosphere, *Q_1_*, and in the H_2_/H_2_O gas mixture, *Q_2_*, is equal to:(1)$$Q_{1}-Q_{2}=-\frac{t_{O}+t_{H}}{2F}\frac{\Delta_{f}H_{H_{2}O}^{{}^{\circ}}}{T}$$
where *t*_O_ and *t*_H_, $\Delta_{f}H_{H_{2}O}^{{}^{\circ}}$ and *F* are transference number of oxide ions and protons, and standard formation enthalpy of gaseous water and Faraday constant, respectively. Therefore, transference numbers of ionic species in BZY10 are required to perform this normalization procedure. In order to calculate them it is necessary to separate contributions of all the charge carriers (their partial conductivities) to the total conductivity of BZY10. At relatively high temperatures, employed in this work, holes and protons are minority defects and, therefore, the charge neutrality condition for BZY10 can be written as $\left[Y_{\mathrm{Zr}}^{/}\right]=2\left[V_{O}^{••}\right]$. As a result, concentrations of holes and protons can be expressed as [21]:(2)p=K·pO214, [OHO•]=K/·pH2O12
where *K* and *K*^/^ are proportionality constants which can be related to the equilibrium constants corresponding to the defect reactions describing the formation of the charge carriers, as discussed in [21]. Since holes were shown in our previous work [21] to exhibit strong trapping by acceptor impurities, most probably by the dopant, $Y_{\mathrm{Zr}}^{/}$, therefore, *p* in the Equation (2) denotes the concentration of ‘free’, non-trapped small polaron holes. These considerations finally result in the following equation for the total conductivity:(3)σ=σO+σp°·pO214+σH°·pH2O12
where $\sigma_{O},\;\sigma_{p}^{{}^{\circ}}$ and $\sigma_{H}^{{}^{\circ}}$ are oxide ion conductivity, hole conductivity at *p*O_2_ = 1 atm and proton conductivity at *p*H_2_O=1 atm, respectively. Each of these conductivities, in turn, is thermally activated and, as a consequence, can be represented using Arrhenius-type equation. This leads to the Equation (4) which can be fitted to the experimental data on total conductivity of BZY10:(4)σ=A·exp(−EA, ORT)+B·exp(−EA, pRT)·pO214+C·exp(−EA, HRT)·pH2O12
where *A*, *B* and *C* are corresponding pre-exponential factors and *E*_A,O_, *E*_A,p_ and *E*_A,H_ are activation energies of oxide ion, hole and proton conduction, respectively. A 4D-fitting procedure was used since total conductivity depends on three independent parameters—*T*, *p*O_2_ and *p*H_2_O, as follows from Equation (4). This allowed fitting all the data on total conductivity of BZY10 simultaneously. The fitting results are summarized in Table 1 and, as an example, are shown in Figure 3 as 3D-plots at two temperatures −1038 and 838 °C, and in Figure 2 as a 2D-plot at log(*p*H_2_O/atm) = −1.73.

Interestingly, the activation energy of proton conduction was fitted as equal to zero within the uncertainty limits, as seen in Table 2. The reason for this is, obviously, two counteracting trends: The increase in thermally activated mobility of protons with temperature and simultaneous decrease in their concentration due to dehydration (see Appendix A). The activation energy of proton mobility in BZY10 was reported to be about 0.44 eV [2,3,36]. Combining this value with the activation energy given in Table 1 allows estimating the hydration enthalpy as about −0.84 eV or −81 kJ·mol^−1^, which is in excellent agreement with values reported in literature [2,3,5,37] and determined by us from the TGA results (see Appendix A). The parameters given in Table 1 allow calculation of partial conductivities and transference numbers of all charge carriers in BZY10. Examples of such calculations are shown in Figure 4 where transference numbers are presented as a function of *p*O_2_ and *p*H_2_O at 1038 °C.

As seen, BZY10 is mainly ionic conductor (with dominating proton conductivity under wet atmosphere and oxide ion conductivity under dry atmosphere) at elevated temperatures under reducing conditions, whereas the hole conductivity dominates under oxidizing conditions. This conclusion is in agreement with similar observations of others [5,6,7].

Using the calculated transference numbers, it is possible to perform the normalization procedure described above for Seebeck coefficient. Thermodynamic data necessary for calculation according to Equation (1) were taken from [38]. The as-normalized Seebeck coefficient is shown in Figure 5 as a function of *p*O_2_ at different temperatures and at log(*p*H_2_O/atm) = −1.73.

Normalized Seebeck coefficient of the proton conductor can be represented [39] as:(5)$$Q=t_{p}\theta_{p}+t_{O}\theta_{O}+t_{H}\theta_{H}+\frac{t_{O}+t_{H}}{4F}\left(S_{O_{2}}^{{}^{\circ}}-R{\mathrm{lnp}}_{O_{2}}\right)+\frac{t_{H}}{2F}\left(-S_{H_{2}O}^{{}^{\circ}}+R{\mathrm{lnp}}_{H_{2}O}\right)$$
where *θ*_p_, *θ*_O_ and *θ*_H_ are partial Seebeck coefficients of holes, oxide ions and protons, respectively. These partial contributions may be defined by the following equation [27,39]:(6)$$\theta_{i}=\frac{1}{z_{i}F}\left(\bar{s_{i}}+\frac{q_{i}^{*}}{T}\right)$$
where $\bar{s_{i}}$ is partial molar entropy of charge carrier and $q_{i}^{*}$—its heat of transport. The calculation of $\bar{s_{i}}$ allows obtaining the following expressions for partial Seebeck coefficients [27,31,39,40]:(7)$$\theta_{p}=\frac{R}{F}\left(\ln\frac{N-p}{p}+\frac{q_{p}^{*}}{RT}\right)$$
(8)$$\theta_{O}=-\frac{R}{2F}\left(\frac{{\bar{s}}_{O\left(vibr\right)}}{R}+\ln\frac{\left[V_{O}^{••}\right]}{3-\left[V_{O}^{••}\right]-\left[OH_{O}^{•}\right]}+\frac{q_{H}^{*}}{RT}\right)$$
(9)$$\theta_{H}=\frac{R}{F}\left(\frac{{\bar{s}}_{H\left(vibr\right)}}{R}+\ln\frac{3-\left[V_{O}^{••}\right]-\left[OH_{O}^{•}\right]}{\left[OH_{O}^{•}\right]}+\frac{q_{O}^{*}}{RT}\right)$$
where *N* is a number of positions available for holes, ${\bar{s}}_{O\left(vibr\right)}$ and ${\bar{s}}_{H\left(vibr\right)}$ are vibrational contributions to the entropy of oxide ions and protons, respectively. According to Tsidilkovsky et al. [39,40], vibrational contribution to the chemical potential of an ionic species (oxide ions or protons) can be estimated in the harmonic approximation as follows:(10)$$\Delta \mu_{i,\left(vibr\right)}=RT\overset{3}{\underset{i=1}{\sum }}\ln\left(2\sinh\left[\frac{\hbar \omega_{i}}{2kT}\right]\right)$$
where *ω_i_*—vibration frequency of proton or oxide ion. In this model, the following set of three frequencies is used for both protons and oxide ions: $\omega_{H}\left({\mathrm{cm}}^{-1}\right)=\left\{3400,\;\;960,\;\;960\right\}$ and $\omega_{O}\left({\mathrm{cm}}^{-1}\right)=\left\{550,\;\;250,\;\;250\right\}$, respectively [39,40]. The vibrational contribution to the entropy of ionic charge carriers can then be calculated as follows:(11)$$\Delta {\bar{s}}_{i,\left(vibr\right)}=-\left(\frac{\partial \Delta \mu_{i,\left(vibr\right)}}{\partial T}\right)=-R\overset{3}{\underset{i=1}{\sum }}\ln\left(2\sinh\left[\frac{\hbar \omega_{i}}{2kT}\right]\right)+RT\overset{3}{\underset{i=1}{\sum }}\frac{\hbar \omega_{i}\cosh\left[\frac{\hbar \omega_{i}}{2kT}\right]}{2kT^{2}}$$

Equations (7)–(11) were substituted into Equation (5). The concentration of holes was taken as p=exp(ΔSp°R−ΔHp°RT)pO214 , where $\Delta S_{p}^{{}^{\circ}}$ and $\Delta H_{p}^{{}^{\circ}}$ are standard entropy and enthalpy of hole formation, respectively. The concentration of protons was calculated from the results of the separate TGA experiment (see Appendix A). Finally, when all the necessary terms were substituted to Equation (5), the Seebeck coefficient was expressed as a function $Q=f\left(T,\;\;pO_{2},\;\;q_{p}^{*},\;\;q_{O}^{*},\;\;q_{H}^{*},\;\;\Delta S_{p}^{{}^{\circ}},\;\;\Delta H_{p}^{{}^{\circ}},\;\;N\right)$. The unknown parameters: $q_{p}^{*}$, $q_{O}^{*}$, $q_{H}^{*}$, $\Delta S_{p}^{{}^{\circ}}$ and $\Delta H_{p}^{{}^{\circ}}$ were determined by fitting Equation (5) to the data on normalized Seebeck coefficient vs. T and *p*O_2_ at fixed log (*p*H_2_O/atm) = –1.73, i.e., a 3D-fitting procedure was employed and all the data were treated simultaneously. The results of the fit are summarized in Table 2 and shown in Figure 5 as 3D- and 2D-plots. It is to be noted that strong correlation between the values of $\Delta S_{p}^{{}^{\circ}}$ and $q_{p}^{*}$ was found during the fitting procedure, therefore, $q_{p}^{*}$ was set equal to zero which seems to be the common assumption in the case of small polarons [28,29,30].

Furthermore, parameters *N* and $K=\exp\left(\frac{\Delta S_{p}^{{}^{\circ}}}{R}-\frac{\Delta H_{p}^{{}^{\circ}}}{RT}\right)$ are also strongly interdependent, since both of them enter in the same part of Equation (7): *N*—in the numerator and *K*—both in the numerator and denominator. As seen in Table 2, fitting leads to different values of $\Delta S_{p}^{{}^{\circ}}$ parameter depending on the particular value of *N* selected. At the same time, the enthalpy $\Delta H_{p}^{{}^{\circ}}$ obtained during the fitting procedure is always the same irrespective of a value of *N*. Therefore, additional assumptions on the particular value of *N* are needed in order to evaluate *K* and then to estimate the concentration of non-trapped holes according to Equation (2). This calculation sequence allows for some speculation on the possible sites available for holes. Indeed, both the charge neutrality condition employed, $\left[Y_{\mathrm{Zr}}^{/}\right]=2\left[V_{O}^{••}\right]$, and excellent quality of fit of Equation (4) to the total conductivity data of BZY10 may imply that holes, either trapped or ‘free’, are minority charge carriers, i.e., that their concentration is expected to be very small. At the same time, fitting Equation (5) under assumption that $N=\left[O_{O}^{\times }\right]=2.95$, which means that holes are associated with oxygen sites, leads to a large positive value of $\Delta S_{p}^{{}^{\circ}}$ parameter (see Table 2) and, in combination with $\Delta H_{p}^{{}^{\circ}}=0.74\;\mathrm{eV}$, to abnormally large value of constant *K*: for example, *K* = 0.092 at 1038 °C. This results in extraordinarily large concentration of non-trapped holes, e.g., *p* = 0.062 at *p*O_2_ = 0.21 atm. A much better agreement with physical meaning is obtained when setting lower values of *N*, for example, at *N* = 0.1 one can obtain *K* = 3.12 × 10^−3^ and *p* = 0.0021 at 1038 °C and *p*O_2_ = 0.21 atm. It may well be that an even significantly lower value of *N* is not unreasonable. Therefore, these simple calculations do not indicate in favor of oxygen sites as a place for localization of holes, contrary to the common belief [25,26,41,42,43,44,45,46]. It seems more likely that the holes are localized on the cation impurities, especially taking into account that significantly different conductivity values were reported for BZY10 by different authors [7,47]. Indeed, different acceptor impurity concentrations may be responsible for such scatter in the reported conductivity values. In order to understand the nature of holes in more detail, it would be of interest to prepare a number of samples with a controlled amount of some selected impurities and then measure and compare their properties. This work is in progress now.

It is also worth mentioning that for the oxide ions the heat of transport is about one half their activation energy of mobility (see Table 1), whereas for the protons these values are almost the same.

## 4. Conclusions

The sample of BZY10 proton conducting electrolyte was prepared and its electrical conductivity and Seebeck coefficient were studied in the wide range of conditions. Conductivity as a function of *p*O_2_, *p*H_2_O and *T* was successfully fitted by the model equation derived on the basis of the defect chemical approach. The fitted model parameters allowed calculation of the transference numbers of all the charge carriers involved. The Seebeck coefficient of BZY10, measured under a reducing atmosphere, was normalized to that measured under oxidative conditions in order to account for the shift in heterogeneous part of thermo-EMF when going from N_2_/O_2_ to H_2_/H_2_O gas mixture. Normalized Seebeck coefficient of BZY10 as a function of *p*O_2_, *p*H_2_O and *T* was fitted by the appropriate model equation containing the weighted sum of partial Seebeck coefficients of each charge carrier and heterogeneous part that depends on the chemical potential of oxygen and water vapor. The heats of transport of oxide ions and protons were found to be about one half of the activation energy of mobility and almost equal to that, respectively. Fitting results of the Seebeck coefficients also revealed that oxygen sites do not seem to be responsible for the formation of small polaron holes, rather, some cation impurities should be considered.

## Figures and Tables

**Figure 1 membranes-09-00120-f001:**
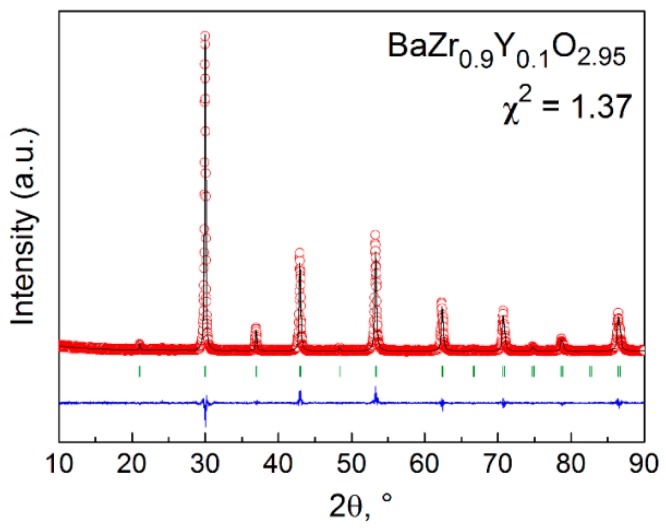
Rietveld refined XRD pattern of BZY10 sample slowly (~100 °C/h) cooled from 1500 °C to room temperature in dry air (log(*p*H_2_O/atm) = −4.0): Observed X-ray diffraction intensity (points) and calculated curve (line). The bottom curve is the difference of patterns, y_obs_-y_cal_, and the small bars indicate the angular positions of the allowed Bragg reflections

**Figure 2 membranes-09-00120-f002:**
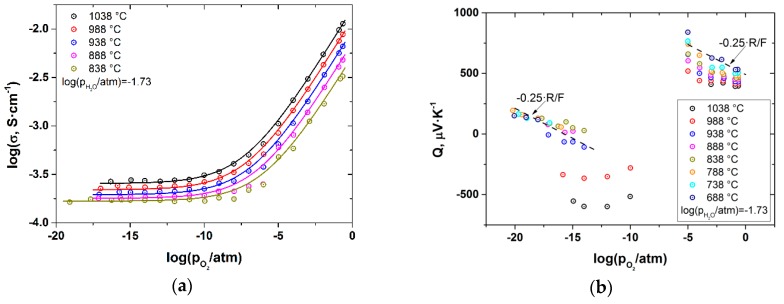
Total conductivity (**a**) and Seebeck coefficient (**b**) of BZY10 measured at log(*p*H_2_O/atm) = –1.73 as a function of *T* and *p*O_2_. Points-experimental results, solid lines-calculation according to Equation (4) with parameters given in Table 1.

**Figure 3 membranes-09-00120-f003:**
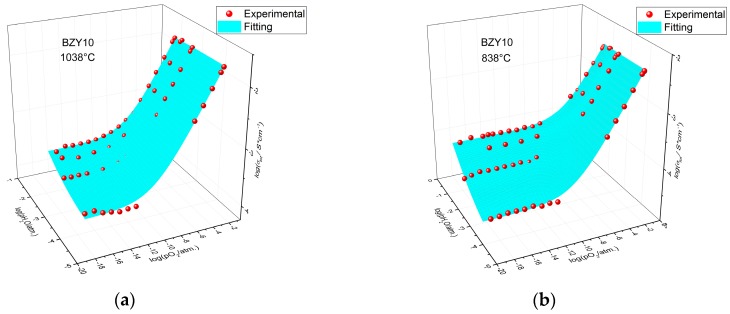
The results of fitting for BZY10 total conductivity at (**a**) 1038 °C and (**b**) 838 °C. Points-experimental data, surface-results of calculation according to Equation (4) with fitted parameters given in Table 1.

**Figure 4 membranes-09-00120-f004:**
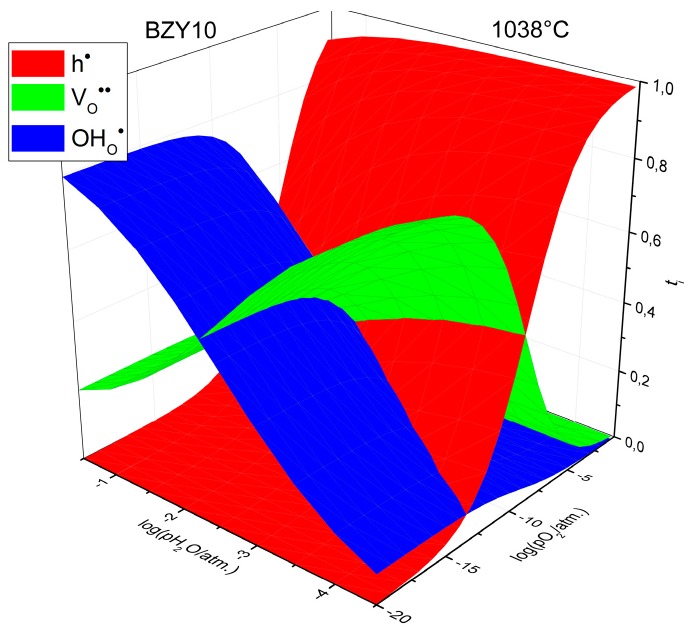
Transference numbers of charge carriers in BZY10 depending on *p*O_2_ and *p*H_2_O at 1038 °C.

**Figure 5 membranes-09-00120-f005:**
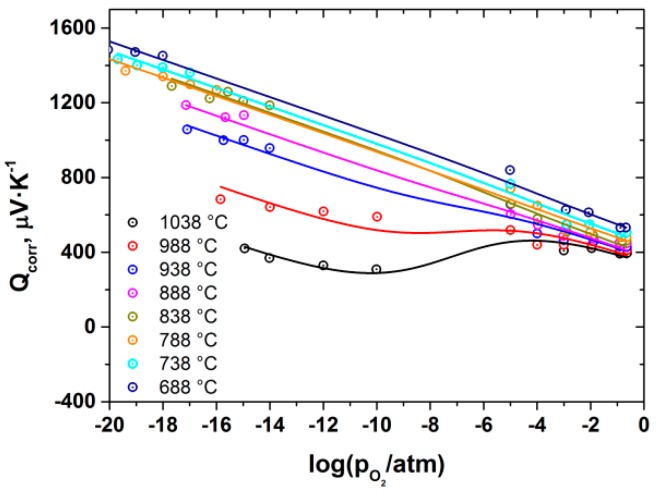
Normalized Seebeck coefficient of BZY10 vs. *p*O_2_ at log(*p*H_2_O/atm) = −1.73. Points-experimental data, lines-result of the fit of Equation (5).

**Table 1 membranes-09-00120-t001:** Fitted parameters of Equation (4)

Charge Carrier	Pre-Exponential Factor ^1^, S·cm^−1^	Activation Energy ^1^, eV	*R* ^2^
Oxide ion	25.43 ± 13.00	1.42 ± 0.01	0.997
Hole	10.42 ± 3.00	0.73 ± 0.05	
Proton	(1.47 ± 0.60)·10^−3^	0.02 ± 0.09	

^1^ Uncertainties are given as two standard deviations as obtained by fitting procedure.

**Table 2 membranes-09-00120-t002:** Fitted parameters of Equation (5).

Charge Carrier	Heat of Transport, eV *	$\Delta S_{p}^{{}^{\circ}},\;J\cdot {\mathrm{mol}}^{-1}\cdot K^{-1*}$	$\Delta H_{p}^{{}^{\circ}},\;\mathrm{kJ}\cdot {\mathrm{mol}}^{-1*}$	*R* ^2^
Oxide ion	0.73 ± 0.03	34.6 ± 1.4 (*N* = 2.95)	0.74 ± 0.04 (does not depend on *N*)	0.990
Hole	0 **	6.5 ± 1.4 (*N* = 0.1)		
Proton	0.43 ± 0.10	−12.3 ± 1.4 (*N* = 0.01)		

* Uncertainties are given as two standard deviations as obtained by fitting procedure.

** $q_{p}^{*}$ was arbitrarily set equal to zero (please, see the explanation in the text)

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
