# Peer review of "Thermoelectric Behavior of BaZr0.9Y0.1O3−d Proton Conducting Electrolyte"

_membranes, 2019, doi:10.3390/membranes9090120_

Round 1

Reviewer 1 Report

This manuscript can be published as it is.

Author Response

Reviewer #1.

This manuscript can be published as it is.

Reply:

We are grateful for such a positive evaluation of our work.

Reviewer 2 Report

Dear Authors,

Thank you for submitting your research work to Memebranes. There are a few questions need to be addressed before publishing. 

1) How many times that you measured the conductivity for each data point? Did you take the mean value?

2) Formulate the errors of seebeck coefficient caused by the measurement error of conductivity and others.

3) Explain the difference caused by the temperature increase to no less than 988C in Figure 2b and Figure 5 in details. Why the trend is different from other cases at lower temperature?

Thank you. 

Author Response

Reviewer #2.

Dear Authors,

Thank you for submitting your research work to Memebranes. There are a few questions need to be addressed before publishing.

Reviewer’s comments:

1) How many times that you measured the conductivity for each data point? Did you take the mean value?

Reply:

Let us describe in more detail the exact measurement procedure that is used for the conductivity measurements in all our works, including the present one. During the measurements of both conductivity and Seebeck coefficient, the sample is allowed to equilibrate at each pO2 and T until its conductivity and Seebeck coefficient ceases to change. This typically requires at least 5-10 hours, but, in order to be sure that the equilibrium state has been reached, the measurement is continued for another 3 hours. During these last 3 hours, the conductivity and Seebeck coefficient are being automatically recorded each second, and in the end all the 180 measurements are averaged. This procedure is repeated at each pO2 and T studied.

Furthermore, all the measurements are carried out, first, in the direction of stepwise decreasing pO2 and then – increasing pO2. The resulting curves, conductivity (or Seebeck coefficient) vs. pO2 at given T, obtained in the forward and backward directions, must coincide completely, provided that the true equilibrium state of the sample has been reached.

2) Formulate the errors of seebeck coefficient caused by the measurement error of conductivity and others.

Reply:

We assume that ‘measurement uncertainty’ as a characteristic of the dispersion of the experimental values is meant by ‘error’, because ‘error’ typically means the deviation of the measured value from the true one, which in our case is not (and cannot be) known exactly. While, for these measurements, the error cannot be estimated, the uncertainty can (and was – see the explanations below).

Let us emphasize that, though conductivity and Seebeck coefficient are measured in the same setup, they are measured absolutely independently of each other. These values are measured alternately. When Seebeck coefficient is being measured, the DC current used for the conductivity measurements is turned off, and vice versa. That is why these measurements neither influence each other nor affect the uncertainties of each other.

As for the other sources of uncertainties, we have already reported the uncertainties in the manuscript (see Section 2. Materials and Methods) as not exceeding ±5% for total conductivity and ±10 μV/K for Seebeck coefficient, so it should be quite enough to judge our experimental data on their quality. The uncertainties were evaluated, according to the NIST classification, as Type B standard uncertainties, corresponding to ca 95% confidence interval. These uncertainties are based on the previous measurements of different samples of the same chemical composition in the same setup, as well as on comparing our measurements with the most reliable of those reported in literature.

 3) Explain the difference caused by the temperature increase to no less than 988C in Figure 2b and Figure 5 in details. Why the trend is different from other cases at lower temperature?

Reply:

Unfortunately, we do not quite understand which particular differences and trends in Figs 2b and 5 caused the Reviewer’s concern. As indicated in the manuscript, the Seebeck coefficient is positive and grows with decreasing pO2 under oxidizing atmosphere, as seen in Fig. 2b. Under reducing conditions (pO2 < 10-5), it drops abruptly, becoming negative, and increases upon subsequent pO2 decrease. The reason for the abrupt change of Seebeck coefficient in the intermediate pO2 range mentioned above is related to the fact that the heterogeneous part of the thermo-EMF is strongly dependent on the chemical potential of oxygen in the surrounding atmosphere. The chemical potential, in turn, is determined by gaseous O2 under oxidizing conditions (10-5≤pO2≤0.21 atm) and by the H2/H2O chemical equilibrium under reducing atmosphere (at pO2 lower then ~10-10 atm). Moreover, with temperature growth, the sample increasingly dehydrates, and proton transference number decreases, whereas that of oxide ions – increases. As a result, proton contribution to the total thermoelectric power decreases and sample behaviour approaches that of the conventional oxide ion conductors. Therefore, the behaviour of the as-measured Seebeck coefficient at low pO2 is quite complex. These considerations, also presented in the manuscript, fully explain the complexity of trends of dependences presented in Fig. 2b.

Then, in order to be able to analyse the sample’s thermoelectric power and to see its overall trend vs. pO2, one needs to normalize the Seebeck coefficient measured in H2/H2O atmosphere against that obtained under O2/N2 atmosphere in order to get rid of the shift in the heterogeneous part of thermo-EMF due to shift in oxygen chemical potential in the surrounding atmosphere when going from O2/N2 to H2/H2O atmosphere. The difference between the values of Seebeck coefficient measured at the same pO2 under O2/N2 atmosphere and in the H2/H2O gas mixture is given by the Eq. (1) (Please, see the manuscript). The as-normalized values of Seebeck coefficient are shown in Fig. 5. Here one can observe gradual variation of Seebeck coefficient with pO2 and T. Under oxidizing conditions the thermopower of the sample is mainly dominated by holes whereas at low pO2 – by protons and oxide ions.

Additionally, it can be pointed out that all the dependences in Fig. 5 were obtained by the same normalization procedure for all temperatures, and that all these dependences can be described by the same theoretical expression (Eq. (5)). All the dependencies, above and below 988 °C, irrespective of the temperature, have essentially the same shape, so there is no real difference in the trends. Seemingly more pronounced shape for the dependencies above 988 °C is owed solely to the interplay between temperature- and pressure-dependent terms in the same Eq. 5.

Reviewer 3 Report

The authors studied the thermoelectric behavior of BZY10 and find that cation impurities should be considered as a place of holes localization. Overall, the study is interesting. The work is publishable after the following issues are addressed.

The authors pressed the powder sample into tablets and did the measurements. The tablets should be characterized, such as surface morphology and roughness, which is important information to understanding the following measurements. Will the pressing and sintering condition influence the results and conclusions in this work? Authors should provide statistical results to show how reliable the measurement is.

Author Response

Reviewer #3.

The authors studied the thermoelectric behavior of BZY10 and find that cation impurities should be considered as a place of holes localization. Overall, the study is interesting. The work is publishable after the following issues are addressed.

The authors pressed the powder sample into tablets and did the measurements. The tablets should be characterized, such as surface morphology and roughness, which is important information to understanding the following measurements. Will the pressing and sintering condition influence the results and conclusions in this work? Authors should provide statistical results to show how reliable the measurement is.

Reply:

As indicated in the manuscript, having been pressed, the green tablets were further sintered at 1500 °C for 24 hours in air with 100 °C/hour as a heating/cooling rate. Before the sintering procedure, green tablets were covered by the sacrificial powder of the same chemical composition in order to prevent Ba loss due to volatility of its oxide at high temperatures. The relatively moderate sintering temperature was chosen because of the same reason. It can be seen that measures were taken to ensure that sintering conditions did not alter the chemical composition of the sample, which, had it been changed during the synthesis, could have affected its conductivity. Rectangular bars of 20×3×3 mm3 for the conductivity and Seebeck coefficient measurements were cut from the tablets sintered accordingly. The as-obtained bars had 85-90% relative density. The residual porosity promoted sample equilibration with the gas phase, as discussed in details by Wang and Virkar (see Ref. [16] in the manuscript).

Unfortunately, we cannot agree with Reviewer concerning the importance of surface morphology and roughness for understanding the particular results presented in the manuscript, since both the morphology of the sample surface and its roughness have no influence on the thermoelectric behavior and bulk conductivity of the sample. What the Reviewer might have had in mind is that pressing and sintering conditions may, indeed, affect the absolute value of the sample’s conductivity through the variation of porosity (or density) of the as-prepared ceramics. However, the sample’s density was actually measured and presented in the manuscript along with detailed description of the sintering conditions, allowing comparing the absolute values of conductivity obtained in our work with the results of others. Moreover, the activation energies of partial conductivities as well as the transference numbers of charge carriers remain unaffected (see Ref. [16] in the manuscript). Seebeck coefficient is also insensitive to porosity and surface morphology. Therefore, the main conclusions of our work remain unchanged irrespective the particular sintering conditions. To reiterate, we firmly believe that no additional surface morphology and roughness measurements are needed for understanding the properties measured in our work.

As the statistical results, showing the reliability of the measurements, some form of uncertainty estimate is used in most cases. We have already reported the uncertainties in the manuscript (see Section 2. Materials and Methods) as not exceeding ±5% for total conductivity and ±10 μV/K for Seebeck coefficient, so it should be quite enough to judge our experimental data on their quality. The uncertainties were evaluated, according to the NIST classification, as Type B standard uncertainties, corresponding to ca 95% confidence interval. These uncertainties are based on the previous measurements of different samples of the same chemical composition in the same setup, as well as on comparing our measurements with the most reliable of those reported in literature. Unfortunately, we do not see which additional statistical results should be presented to facilitate the understanding of the reliability of our measurements.

During the measurement procedure the sample is allowed to equilibrate at each pO2 and T until its conductivity and Seebeck coefficient ceases to change. This typically requires at least 5-10 hours but in order to be sure that the equilibrium state was reached the measurement is continued for another 3 hours. During these last 3 hours the conductivity or Seebeck coefficient is being automatically measured each second and in the end all the 600 measurements are averaged. This procedure is repeated at each pO2 and T studied. Furthermore, all the measurements are carried out, first, in the direction of stepwise decreasing pO2 and then increasing pO2. The resulting curves, conductivity (or Seebeck coefficient) vs. pO2 at given T, obtained in the forward and backward directions must coincide completely provided that the true equilibrium state of the sample was reached.

Round 2

Reviewer 3 Report

The revised manuscript is now publishable.